# The Role of Complement System and the Immune Response to Tuberculosis Infection

**DOI:** 10.3390/medicina57020084

**Published:** 2021-01-20

**Authors:** Heena Jagatia, Anthony G. Tsolaki

**Affiliations:** 1Department for Respiratory Sciences, University of Leicester, Leicester LE1 9HN, UK; 2Department of Life Sciences, College of Health and Life Sciences, Brunel University of London, Uxbridge UB8 3PN, UK; anthony.tsolaki@brunel.ac.uk

**Keywords:** macrophages, tuberculosis, granuloma, complement, chronic inflammation

## Abstract

The complement system orchestrates a multi-faceted immune response to the invading pathogen, *Mycobacterium tuberculosis*. Macrophages engulf the mycobacterial bacilli through bacterial cell surface proteins or secrete proteins, which activate the complement pathway. The classical pathway is activated by C1q, which binds to antibody antigen complexes. While the alternative pathway is constitutively active and regulated by properdin, the direct interaction of properdin is capable of complement activation. The lectin-binding pathway is activated in response to bacterial cell surface carbohydrates such as mannose, fucose, and *N*-acetyl-*d*-glucosamine. All three pathways contribute to mounting an immune response for the clearance of mycobacteria. However, the bacilli can reside, persist, and evade clearance by the immune system once inside the macrophages using a number of mechanisms. The immune system can compartmentalise the infection into a granulomatous structure, which contains heterogenous sub-populations of *M. tuberculosis*. The granuloma consists of many types of immune cells, which aim to clear and contain the infection whilst sacrificing the affected host tissue. The full extent of the involvement of the complement system during infection with *M. tuberculosis* is not fully understood. Therefore, we reviewed the available literature on *M. tuberculosis* and other mycobacterial literature to understand the contribution of the complement system during infection.

## 1. Introduction

*Mycobacterium tuberculosis* was first discovered by the German physician and microbiologist Robert Koch in 1882, who subsequently won the Physiology or Medicine Nobel prize in 1905 (nobelprize.org). The *M. tuberculosis* cells are acid-fast rods and obligate aerobes with a doubling time of 22 h [1]. *M. tuberculosis* is the causative agent of the chronic disease, tuberculosis, which caused 10 million new cases and 1.5 million deaths in 2018 [2]. *M. tuberculosis* was classified as one of the top ten causes of deaths worldwide in 2016, and importantly, one-quarter of the worlds’ population is infected with latent tuberculosis [2].Tuberculosis is primarily transmitted through the inhalation of infected air droplets and symptoms include night sweats, coughing blood, fatigue, and fever. Despite public misconceptions, tuberculosis is not an eradicated disease in the UK; there were 4725 reported cases in England in 2019 [3]. The complex bacterial cell wall, composed of arabinogalactan and mycolic acids, has proven to be difficult to penetrate therapeutically, which goes some way to explaining the continued difficulty in eradicating tuberculosis. Other barriers that have contributed to the problems with tuberculosis eradication include drug development failures and a rise of drug resistance. Treatment for drug susceptible tuberculosis comprises of a combination of four drugs over a 6-month period, whilst drug resistant infections require up to 24 months of treatment.

## 2. Widely Accepted Paradigm of Infection

The agreed paradigm of infection with *M. tuberculosis* can be separated into early and late infection. The early acute phase begins with the inhalation of infected airborne droplets from an infected patient. The bacilli enter the alveolar space and are phagocytosed by the abundant resident macrophages, which are favoured over any other cell type [4]. Type II alveolar epithelial pneumocytes have been shown to allow the internalisation of *M. tuberculosis* in vitro [5]. These macrophages internalise *M. tuberculosis* through bacterial cell surface proteins or through secreted proteins that activate the complement pathway [6]. The complement component, C3, is present in serum and opsonises the surface of *M. tuberculosis* for recognition by macrophages through the complement receptor [6]. The absence of serum results in the recognition of *M. tuberculosis* cell surface ligands, lipoarabinomannan (LAM), via the mannose receptors; however, this scenario has been reported for virulent strains of *M. tuberculosis* [6]. The bacilli are capable of evading host protective mechanisms which includes phagosome–lysosome fusion, recruitment of hydrolytic lysosomal enzymes, production of reactive oxygen and nitrosative species, apoptosis, and evasion of antigen presentation, once the bacilli have entered the phagosomal compartment of macrophages [7].

The chronic phase of infection involves the survival of *M. tuberculosis* in the macrophages and is referred to as the latent infection; factors that contribute to this can be host and pathogen derived, for example, the virulency of *M. tuberculosis* and host immune status. The failure of macrophages to eliminate *M. tuberculosis* infection results in the recruitment of monocytes, which differentiate into macrophages. This causes the further phagocytosis of *M. tuberculosis* in a passive manner, and the cycle continues, causing the logarithmic growth of tuberculosis infection within the macrophages. The production of chemokines recruits’ lymphocytes and neutrophils to the site of infection, which are not capable of clearing infection but increase the inflammatory effects. Eventually, macrophage-derived giant cells and lymphocytes cause the development of a caseous granuloma to compartmentalisation of the infection; however, the exact factors that drive granulomatous formation are not fully understood [8]. The bacilli move into the lung interstitial space, which is rich in complement proteins, and further drives local inflammation, which consequently drives the influx of more macrophages and neutrophils from the blood and dendritic cells from the lung parenchyma [9,10]. Tissue-resident dendritic cells play a key role in *M. tuberculosis* antigen presentation and the development of anti-mycobacterial T cell responses after a couple of weeks of infection [11]. The T cells produce interferon-γ (IFN-γ), which activates the macrophages to kill and eliminate infection and thus causes the halt of the logarithmic growth of *M. tuberculosis* [12].

The role of the complement system in *M. tuberculosis* infection is poorly understood. There are a handful of publications that show complement pathway activation during challenge with *M. tuberculosis*; however, there is very little understanding of the exact involvement of the complement cascade and its contribution to tuberculosis disease progression. Therefore, this review focuses on the evidence for the role of the complement system during *M. tuberculosis* infection.

## 3. Classical Pathway and Mycobacteria

C1q activates the classical pathway, which is the first subcomponent of the classical complement cascade. C1q is a hexameric molecule (460 kDa) with a characteristic tulip-like structure [13] that binds directly to the surface of bacteria or can be activated during the adaptive immune response by binding to the antibody–antigen complexes. The binding of C1q results in the cleavage of C4 and C2 to the convertase C4bC2a. Then, this convertase cleaves C3 to produce C3a and C3b, which are ligands that opsonise *M. tuberculosis.* Most individuals received the Bacillus Calmette-Buérin (BCG) vaccination in the UK up until 2005, so studies in the model organism, *Mycobacterium bovis* BCG, have shown C1q binding in the presence of Immunoglobulin G (IgG) and immune globulin M (IgM) in serum samples [14]. The presence of anti-lipoarabinomannan (LAM) IgG2 in the sera of Indian tuberculosis patients has correlated with classical activation by BCG in sera [15]. C1q-deficient serum has shown a reduction C3 binding [16]. Some argue that the classical pathway is a lot more active than the alternative pathway in the lungs despite its lower abundance of serum, which has been demonstrated through bronchoalveolar lavage fluid [16]. Additionally, Ferguson and colleagues tested the binding of C3 to *M. tuberculosis* and showed that incubating nonimmune human serum for less than 5 min with *M. tuberculosis* resulted in the cleavage of C3 to C3b, C3bi, and other fragments [16]. The antigen–antibody complexes or opsonised bacterial surfaces interact with complement receptors (CR). Some of the early evidence from the 1990s showed that the monocyte complement receptors CR1 and CR3 were responsible for the adherence to the major ligand C3 and ingestion of *M. tuberculosis* [17]. Neutrophils challenged with *Mycobacterium kansasi* showed evidence for CR3 binding [18]. Hu and colleagues investigated the contribution of the CR3, which recognises the iC3b fragment in mycobacteria by using CR3-deficient mice. There was no difference in mortality between CR3^WT^ and CR3^−/−^ mice upon infection with *M. tuberculosis*, nor was there a difference of bacterial burden in the liver, spleen, and lungs, which was confirmed histologically [19]. Others have used *Mycobacterium avium* in CR3^−/−^ mice and have shown that there is no difference in bacterial burden, susceptibility to *M. avium*, and granulomatous response compared to the control mice [20]. However, there was significantly higher invasion of *M. avium* in bone marrow-derived macrophages (BMDM) in CR3^−/−^ mice [20]. The cell surface of *M. avium* was sufficient for activation of the classical complement pathway independent of antibodies in vitro; the levels of C3 cleavage were measured by incubation of C1, C2, C3, and C4 with *M. avium* [20]. The classical pathway and its contribution to the host response during tuberculosis infection is summarized in Figure 1.

Patients with tuberculoid leprosy (TT) exhibit little bacilli in the skin lesions whilst lepromatous leprosy (LL) patients have greater numbers of bacilli in many skin lesions. Leprosy is associated with the development of pathologic immune reactions, reversal reaction (RR), or erythema nodosum leprosum (ENL). Transcriptomic analysis of PBMCs in RR and ENL has shown increased C1q activity, but the C1q levels in blood did not increase [21,22]. This suggests that the C1q was deposited in tissues and therefore able to bind to immunoglobulins to form immune complexes [23]. Those with LL reactions presented with lower levels of C4 and increased with the development of RR or ENL. Polymorphisms in CR1 have been shown to increase susceptibility to *Mycobacterium leprae* [24].

**Figure 1 medicina-57-00084-f001:**
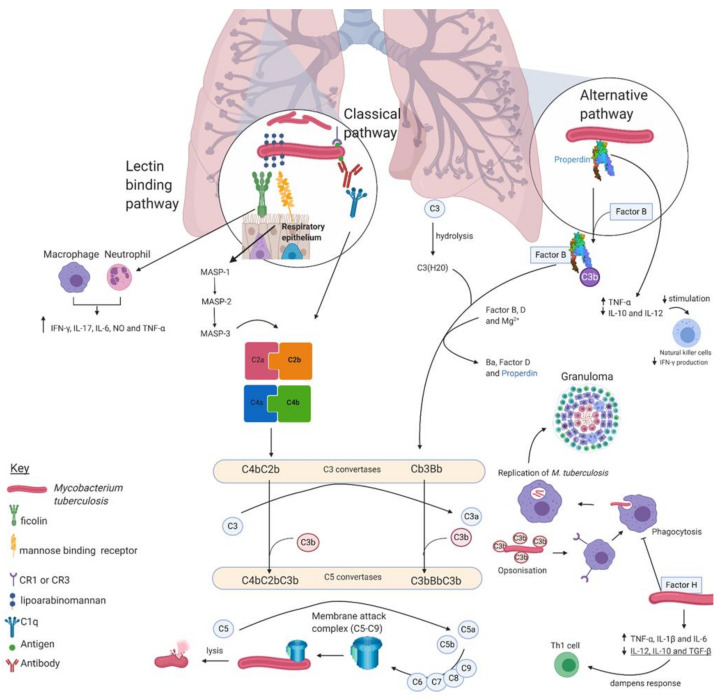
Overview of the complement system in mycobacteria.

A simplified cartoon representing the involvement of the three complement pathways in tuberculosis infection: classical, lectin binding, and alternative. Briefly, the classical pathway is initiated by C1q, which recognises antibody—*M. tuberculosis* antigen complexes. The lectin binding pathway is activated by the recognition of complex sugar moieties on the *M. tuberculosis* cell surface. The alternative pathway is constituently active at basal levels, and unlike the other pathways, it does not require activation. The alternative pathway is regulated by properdin and factor H. All three complement pathways lead to the activation of C3 convertases, which cleaves C3 to C3b. C3b opsonises *M. tuberculosis*, which is ready for recognition by macrophages. The bacilli reside within the macrophage and experience a multitude of stresses induced by the host to clear the infection. This event develops into the characteristic granuloma. The C3 convertases also activate C5 convertases, resulting in the cleavage of C5 to C5a and C5b, leading to the formation of the MAC and ultimately, mycobacterial cell lysis.

## 4. Alternative Pathway and Mycobacteria

Unlike the classical and lectin complement pathways, which require stimuli for activation, the alternative pathway is constitutively active at basal levels, and it is ready to recognise invading pathogens rapidly. The alternative pathway and its contribution to the host response during tuberculosis infection is summarized in Figure 1. The alternative pathway is activated by C3 (C3bBb) convertase, which is bound to the complement-activating target and causes the cleavage of C3. The C3bBb convertase is formed from the association of factor B in the presence of Mg^2+^ and factor D. This causes the cleavage of factor B into Bb and Ba fragments, thus forming the C3bBb convertase. The C3bBb convertase produces the opsonin C3b, which is deposited alongside C3bBb convertase, resulting in the formation of C5 convertase, which produces C5a and C5b fragments and results in cell lysis when the membrane attack complex (C5-9) is inserted in a membrane [25,26].

Properdin is a regulator of the alternative pathway and composed of seven thrombospondin type 1 repeats (TSR1-TSR6). TSR4 is crucial for the stabilisation of C3bBb, and TSR5 contributes to the binding of C3b demonstrated through domain deletion studies [27]. Properdin binds to the bacterial cell surface directly and is responsible for the recruitment of C3b, which causes assembly of the C3 convertase and ultimately more C3b deposition on the bacterial surface. TSR4+5 upregulates tissue necrotic factor-α (TNF-α), which is involved in the growth and maintenance of the granuloma, but it downregulates interleukin 10 (IL-10) and interleukin 12 (IL-12); this subsequently reduces stimulation of natural killer (NK) cells and IFN-γ. Thus, properdin is involved in modulation of the immune response to mycobacterial infection [28]. Unlike properdin, Factor H has been shown to downregulate complement activation by specifically binding to *Mycobacterium bovis* BCG and thus preventing uptake of *M. bovis* BCG by THP-1 macrophages [14,28]. Additionally, Factor H results in the upregulation of TNF-α, interleukin 1 β (IL-1β), and interleukin 6 (IL-6) expression during early phagocytosis. IL-1β provides resistance to mycobacteria, whilst IL-6 enables a T-cell response to *M. tuberculosis* infection. IL-12, IL-10, and tissue growth factor β (TGF-β) are downregulated by factor H, which suppresses the Th1 response and IFN-γ, thus dampening the anti-inflammatory cytokine response [29,30].

## 5. Collectins, Lectin Pathway, and Mycobacteria

An important component of the innate immune system is collectins, which are key pattern-recognition molecules (PRMs) that bind to invading pathogens and contribute to neutralisation and clearance [31]. The role of collectins and the lectin pathway during tuberculosis infection is summarized in Figure 1.

Collectins are also important in influencing the subsequent adaptive immune response against these microbial infections. Among the collectins are surfactant proteins A (SP-A) and D (SP-D), mannose-binding lectin (MBL), liver collectin (CL-L1), kidney collectin (CL-K1), CL-LK (composed of CL-L1 and Cl-K1), and placenta collectin (CL-P1) [32]. Additional collectins have also been described in Bovidae species, which are called conglutinin, CL-43 and CL-46 [32]. Collectins can clear microbial infection through aggregation, opsonisation, inhibition of phagocytosis, and microbial growth inhibition [32]. Of these collectins, MBL, CL-LK, and SP-A have also been shown to be involved in complement interactions.

As discussed, MBL is a major component of the complement system’s lectin pathway and is mainly a serum protein. The structure of MBL is similar to C1q and SP-A and can target terminal sugars on the surface of bacteria in a calcium-dependent manner [33]. SP-A and MBL have globular C-terminal lectin domains that target microorganisms in this manner [32]. This MBL binding to the surface of microbes can subsequently activate complement via MBL-associated serine proteases (MASPs), consequently leading to greater microbial clearance through opsonisation with C3 and C4 components and complement-mediated phagocytosis [32]. Additionally, MBL has complement-independent properties and can inhibit bacterial growth directly and acts as an opsonin in its own right [34,35,36]. MBL can bind several Gram-positive and Gram-negative bacteria, including *Staphylococcus aureus*, *Streptococcus pyogenes*, *Listeria monocytogenes*, and non-encapsulated *Neisseria meningitidis, Escherichia coli, Haemophilus influenzae*, *Klebsiella* spp, and mycobacteria species [37,38,39]. In mycobacteria, MBL has been shown to bind to lipoarabinomannan (LAM) from *Mycobacterium avium* [36] and mannosylated lipoarabinomannan (ManLAM) from several others, e.g., *M. tuberculosis*, *M. bovis*, *M. kansasii*, *Mycobacterium gordonae* and *Mycobacterium smegmatis* [40]. The persistence of *M. leprae* LAM in mice contributes to the deposition of membrane attack complex (MAC), resulting in myelin loss and axonal damage [41]. Conversely, the inhibition of MAC provides neuroprotection in response to *M. leprae* infection [41]. Multi-drug therapy for the treatment of *M. leprae* was believed to release dead bacilli, and *Mycobacterium. leprae* LAM can sustain persistent complement activation, resulting in chronic inflammation [41,42].

Other mycobacterial ligands that can be targeted by MBL include the antigen 85 (Ag85) complex of *M. tuberculosis* [43]. Experiments using the vaccine strain *Mycobacterium bovis* BCG have demonstrated complement activation of all three pathways [14]. For the lectin pathway, this study demonstrated the direct binding of both MBL and L-ficolin from human serum to mycobacteria and subsequent MASP2 activation, resulting in the cleavage of C2 and C4, complement activation, and C3 deposition [14]. Patients with tuberculosis have been reported to have lower ficolin-2 levels compared to healthy controls, and in vitro experiments have shown L-ficolin to bind to virulent mycobacteria more so than non-virulent mycobacteria [44]. Interaction between L-ficolin with neutrophils and macrophages was shown to stimulate IFN-γ, interleukin-17 (IL-17), IL-6, NO, and TNF-α [44]. Presumably, mycobacteria encounter MBL in the serum under situations such as disseminated disease (e.g., extra pulmonary tuberculosis), although the precise role played by MBL in complement activation is not well understood. However, the lectin pathway’s role in overall complement activation in the blood is thought to be less than the classical and alternative pathways, which is probably because of the low levels of initiating its initiating proteins, e.g., MBL and ficolins [14]. It is also worth noting that in other anatomical sites other than the bloodstream, lectin pathway activation against mycobacteria may have more of a role depending on the relative concentrations of C1q, MBL, and ficolins in different bodily fluids, which are currently not well understood [14].

Deficiency in MBL is known to increase susceptibility to microbial infection, despite most individuals being apparently healthy [45,46,47]. MBL serum levels in the human population vary greatly (from 5 to 10,000 ng/mL), and this could be from genetic polymorphisms in the MBL genes [48]. Deficiency in MBL is common, with a quarter of Caucasians observed to have low levels (<500 ng/mL) of MBL, making them more susceptible to infection [49]. In mice, MBL deficiency was observed to make mice more susceptible to *Staphylococcus aureus* infection [50], whilst in humans with severe burns, MBL deficiency was associated with increased susceptibility to infection with *Pseudomonas aeruginosa* [51]. MBL deficiency is also linked to meningococcal infection and pneumococcal pneumonia cancer chemotherapy patients [52,53,54]. In tuberculosis, serum levels of MBL and its association to disease has been studied. There is evidence that both normal or increased levels of MBL are associated with frequent mycobacteria infection with *M. tuberculosis* and *M. leprae* [55,56], which is postulated to occur through enhanced complement-mediated uptake of the pathogen into host cells. Approximately 5 to 30% of healthy humans possess mutations associated to MBL deficiency, which been linked with susceptibility to tuberculosis and other inflammatory diseases in some ethnic minorities [57,58,59]. In contrast, polymorphisms in the MBL gene were associated with protection against tuberculosis meningitis in children [60], showing that genetic changes can be both potentially protective as well as harmful. Additionally, polymorphisms in MBL and MASP-2 genes were associated with the susceptibility of tuberculosis, showing possible gene–gene interactions [61]. However, in 168 patients with tuberculosis, no association was found between ficolin-2, ficolin-3, MASP-2 genotypes, the lectin complement pathway, or serum levels, and susceptibility to pulmonary tuberculosis suggests that high MBL serum levels in these tuberculosis patients were probably due to an acute phase response instead [62].

The serum collectin CL-LK (composed of CL-L1 and CL-K1) can also activate the lectin pathway by via MASP-1 and MASP-2 [63]. The serum levels of CL-LK are lower than those of MBL (approximately 0.3 μg/mL), so CL-LK probably plays a minimal role in pathogen recognition and clearance. However, CL-LK has been reported to bind to *M. tuberculosis* via ManLAM but not to *M. smegmatis* [64]. Furthermore, mice deficient in CL-K1 showed no change in susceptibility to *M. tuberculosis* infection, whilst CL-LK opsonised *M. tuberculosis* does not alter its phagocytosis or intracellular persistence in human macrophages [64]. Intriguingly, serum levels of CL-LK in tuberculosis patients is decreased, compared to controls, perhaps showing its possible utility as a biomarker for tuberculosis [64].

Conglutinin is a type of collectin largely found in bovine serum and is secreted from the liver, binding to iC3b. Truncated conglutinin (rfBC) exhibited bacteriostatic activity in culture and showed an inhibition of phagocytosis of *M. bovis* BCG in THP-1 macrophages [65]. The inhibition of phagocytosis occurs possibly because rfBC interferes with mannose receptor-mediated uptake by masking lipoarabinomannan on the *M. bovis* BCG surface or by preventing an adaptive immune response [65]. Maltose (an analogue of mannose) was shown to inhibit interaction between *M. bovis* BCG and conglutinin and thus indicated competitive inhibition by binding to carbohydrate surface motifs [65]. The incubation of *M. bovis* BCG with complement from human serum caused the inhibition of *M. bovis* BCG with conglutinin-mediated phagocytosis [65].

Lung surfactant proteins SP-A and SP-D can also both bind to Gram-negative and Gram-positive bacteria, aggregating, neutralising, inhibiting their growth, and acting as opsonins [32]. SP-A has structural similarities to C1q and MBL that resemble a bouquet of flowers [66] and has also been shown to regulate complement activation [67]. Furthermore, the receptor for C1q (CD93/C1qRp/C1qR) is also a phagocytic receptor for SP-A and MBL [68]. SP-A can also bind directly to C1q [69], and SP-A may regulate complement activity in the lung as a result. This occurs through two mechanisms: Firstly, SP-A prevents C1q from combining with C1r and C1s to form the complete C1 complex that is needed to activate complement, and secondly, SP-A interferes with immune complex recognition by both C1q and C1 [67]. It is not clear what impact this has on immune defence against microbes in the lung, but these findings suggest that SP-A may be protective against lung inflammation by inhibiting C1q-mediated complement activation. The implications for mycobacterial infection are unknown, but it is well established that SP-A interacts directly with *M. tuberculosis* via its putative adhesin Apa glycoprotein [70]. SP-A, as an opsonin, enhances the phagocytosis of *M. tuberculosis* and *M. avium* by macrophages via the increased expression of mannose receptor [52,71,72], whilst similarly, SP-A can enhance the uptake of *M. bovis* BCG via the specific 210-kDa SP-A receptor (SPR210) in U937 macrophages and rat alveolar macrophages [73,74], leading to enhanced mycobacterial killing and the production of TNF-α and nitric oxide [75].

In humans, there are two isotypes of SP-A (SP-A1 and SP-A2) that are coded for by separate genes [66]. Several mutations at the SP-A gene locus have been associated with both protection from or susceptibility to tuberculosis in human populations around the world (e.g., Mexico, Ethiopia, India and China) [76,77,78,79,80]. These findings demonstrate the complex nature of the host–pathogen interaction of mycobacterial infections mediated by complement and collectins.

## 6. Granuloma Formation and the Complement System

One of the hallmarks of a successful tuberculosis infection is the presence of granulomas, which have been described as solid host-protective structures. However, this idea has vastly changed as a result of the advancement in microscopic technologies that show granulomas to be dynamic structures that allow cells to move in and out [81]. The granuloma is made up of giant cells, which arise from the fusion of mature macrophages and foam cells that are characterised by lipid accumulation [82]. An overview of the granuloma structure has been summarized in Figure 2.

This is an overly simplified cartoon of the complex *M. tuberculosis* granuloma. Briefly, the polarisation of macrophages into M1 and M2 is balanced finely by the host to manage chronic infection. These groups of macrophages are distinct; M1 macrophages promote a pro-inflammatory response using glycolytic respiration, whilst the M2 macrophages promote an anti-inflammatory response. The inflammatory markers regulate the promotion and formation of the granuloma and determines disease outcome.

Foamy macrophages are distinct to non-foamy macrophages as they express higher levels of MHCII, CD11b^+^, CD11c, and CD40; however, they are poor antigen-processing cells [83]. Infected foamy macrophages are packed with host lipid, which is consumed by *M. tuberculosis* through aerobic glycolysis and the induction of genes encoding lipid transporters and lipolytic enzymes [82,84,85]. Live cell imaging studies have shown the localisation and migration of *M. tuberculosis* towards the host lipid reservoirs [86]. Finally, as the foamy macrophages leave the original granuloma, dissemination results in secondary granuloma formation.

Other aspects of the granuloma include necrotic areas also known as caseum, made up of neutrophils, natural killer (NK) cells, dendritic cells (DC), B and T cells. This structure is lined with epithelial cells, histocytes (mature monocytes), and lymphocytes, which play a key role in the granulomatous formation. The histiocytes are activated by T lymphocytes, which secrete IFN-γ, infected macrophages become necrosed (Ghon focus), and thus persistence of chronic infection occurs [43,44,87]. The primary lesion can spread to the local lymph system undergoing latency and calcifications (Ghon complex) [87].

The interrelationships between macrophages and *M. tuberculosis* have an effect on dampening the immune response and cause the increased survival of mycobacteria. Distinct types of macrophages, M1 and M2, can allow permissive or restrictive mycobacterial growth [84,88] M1 macrophages are involved in the pro-inflammatory Th1 response, which is also known as the Warburg effect during the early infection phase. M1 macrophages are stimulated by Toll-like receptor (TLR) ligands and IFN-γ via signal transducer and activator of transcription 1 (STAT1) signaling. IFN-γ results in STAT1 phosphorylation, which upregulates the transcription of nitric oxide synthase; thus, nitric oxide (NO) and NO-derived species are produced [89,90,91]. TNF-α provides protection against *M. tuberculosis* infection by binding to the TNF receptor, which results in the activation of the nuclear factor kappa-light change enhancer of activated B cells (NFKB pathway, and it has been associated with granulomatous formation. Mice that are deficient in TNF-α or TNF-α receptors have been shown to be unable to generate granulomatous structures in response to *M. tuberculosis* [92]. M1 macrophages utilise aerobic glycolysis upon activation for the production of ATP [93,94]. The electron transport chain is also altered during this period for the production of reactive oxygen species (ROS) and NO.

In contrast, M1 macrophages are transformed into M2 macrophages by the 6-kDa early secretory antigenic target (ESAT-6) [91]. *M. tuberculosis* can subvert the switch from M2 to M1 and thus persist and survive in the internal macrophage environment [88,95,96]. M2 macrophages are involved in an anti-inflammatory T helper 2 (Th2) response, which is stimulated by interleukin 4 (IL-4), interleukin 13 (IL-13), and IL-10 via STAT6 and STAT3 signaling [90,97,98,99]. IL-4 is produced in response to *M. bovis* BCG and *M. tuberculosis* lipoarabinomannan from bone marrow cells and induces oxidative phosphorylation and mitochondrial respiration [94,100]. M2 macrophages favour fatty acid oxidation and oxidative metabolism, which continues to generate energy for a longer period [93,94]. Blocking oxidative metabolism can prevent M2 polarising and drives towards M1 [93]. Additionally, a study reported pleural macrophages after combination therapy: isoniazid, rifampicin, pyrazinamide, and ethambutol (HRZE) favoured polarisation to M2-like phenotype [101]. *M. tuberculosis* sub-populations in the granuloma experience hypoxia, causing the pH of the caseous core of the granuloma to range between 7.2 and 7.5. In vitro analysis has shown peripheral blood mononuclear cells switching from M1 to M2 on the induction of hypoxic conditions [102]. Each granuloma is an isolated entity that determines whether it initiates active disease or remains quiescent, and the relative ratio of M1/M2 macrophages can determine the granuloma status [103]. Interestingly, M2 macrophages have been found to be abundant in the lung tissue of MDR-TB patients.

## 7. Th1/Th2 Switch in the Granuloma and the Involvement of Complement System

The maintenance of the granuloma in critical in the control of tuberculosis, and the early interactions of complement and other innate immune factors (e.g., collectins) may be essential in modulating the subsequent adaptive immune response and formation of the granuloma. The granuloma is a complex lesion made up of various immune cells that effectively cordons off *M. tuberculosis* infection and limits its growth and spread, but conversely also is a place for the mycobacterium to persist as latent infection [104]. Inflammatory cytokines such as TNF-α and IFN-γ are critical for granuloma development and are secreted by infected *M. tuberculosis*-infected macrophages early on in the infection process, accelerating the recruitment of cells in the granuloma [105]. Recent studies have suggested that the complement regulatory proteins Factor H and properdin may also play a role in this process by enhancing the pro-inflammatory cytokine responses required for granuloma formation and maintenance [28,29]. Factor H bound to *M. bovis* BCG was found to inhibit phagocytosis by macrophages but also boosted pro-inflammatory cytokines (TNF-α, IL-1β, and IL-6) whilst simultaneously decreasing anti-inflammatory cytokines (IL-10, TGF-β, and IL-12) [29]. Similar findings were also observed with properdin-bound *M. bovis* BCG [28]. It has been suggested that this may be a strategy to limit intracellular infection by *M. tuberculosis* whilst promoting inflammatory conditions for the granuloma and limiting the spread of infection [29]. A balance of inflammation, particularly in the lungs, must be tightly controlled, and this balance is essential for the maintenance of a protective homeostatic granuloma. This balance is referred to as the Th1/Th2 balance and is governed by the relative concentrations of IFN-γ/TNF-α (Th1) versus IL-4/IL-10/TGF-β (Th2) within the granuloma. It is possible that innate immune molecules such as factor H, properdin, C1q, and collectins may influence the early host–pathogen interactions described above for *M. tuberculosis* and influence the inflammatory response, shaping the adaptive immunity (granuloma formation and maintenance through the balance of Th1/Th2). Indeed, Factor H expression is enhanced in monocytes by IFN-γ, and macrophages can locally produce complement proteins such as C1q, suggesting that they may reside within the granuloma [106]. However, it is currently unknown if any complement proteins reside within the granuloma, but they may play a role in preventing Th1/Th2 imbalance, which ultimately results in granuloma necrosis as well as *M. tuberculosis* reactivation and growth, which is also advanced by CD4+ T cell activation [107]. Moreover, the possible suppression of IL-12 by complement proteins, the subsequent reduction in the phagocytosis of mycobacteria, and impaired antigen presentation would lead to a downregulation in the Th1 response and the recruitment of CD4+ T cells secreting IFN-γ. This would be also important in the Th1/Th2 balance within the protective granuloma, where T-cell activation can initiate granuloma necrosis [108].

The suppression of IL-10 and TGF-β may further promote mycobacterial clearance during the early stages of infection rather than promoting *M. tuberculosis* growth [109]. Indeed, IL-10 inhibits phagolysosomal maturation, thus facilitating *M. tuberculosis* persistence [30], and it also inhibits antigen presentation during phagocytosis via major histocompatibility complex (MHC) [110]. Furthermore, IL-10 is involved in suppressing dendritic cell activation [111], leading to a weakened Th1 response with increased mycobacterial growth [112]. The levels of IL-10 and TGF-β in the lungs of active tuberculosis patients is also significantly elevated, suggesting a suppression of the immune response to *M. tuberculosis*, facilitating pathogenesis and disease progression [113]. Thus, the involvement of innate immune molecules such as complement proteins in the phagocytosis of mycobacteria may help diminish the evasion mechanisms employed by *M. tuberculosis*, thus enhancing bacteria clearance in the early stages of infection and facilitating a protective response immune response in the form of a granuloma.

Additionally, studies in zebrafish infected with *M. marinum* have shown that a virulence-associated RD1 locus plays a role in the macrophage internal environment for the development of granuloma formation [114]. RD1 locus knock-outs in zebrafish were reported to dysregulate macrophage aggregation in granulomas and promoted macrophage cell death [114]. Others have used the zebrafish model to study *M. tuberculosis* granulomas, which were subsequently dissected and cultured ex vivo; these granulomas displayed the same heterogeneity in immune cells and in immune response [115].

Trehalose 6,6′-dimycolate (TDM) is a glycolipid component of the mycobacterial cell wall and has been shown to mimic *M. tuberculosis* granulomatous formation in mice [116,117,118]. Complement C5 is secreted directly from the *M. tuberculosis* infected macrophages, and C5a is responsible for the transcription/translation of cytokines. Animal studies have demonstrated that A/J (C5 deficient) and C5aR^−/−^ mice succumb to *M. tuberculosis* infection compared to C57BL/6 mice when treated with TDM from a failure to contain and localise the infection in granulomas [117,118,119]. The BMDM from the A/J mice ex vivo were unable to clear *M. tuberculosis* infection and enhanced the multiplication of *M. tuberculosis* within the macrophages [117,118,119].

Additionally, complement protein 7 (C7) inserts into the cell membrane and plays a vital role in the formation of the membrane attack complex, C5b-9, resulting in cell lysis [120]. A study has shown that C7^−/−^ mice decreased *M. tuberculosis* dissemination into livers, reducing lung immunopathology and smaller granulomas [120]. However, the true effect of C7 in *M. tuberculosis* remains mostly unclear.

The recognition of *M. tuberculosis* by myeloid cells via PRRs causes the activation of nucleotide-binding oligomerisation domain-like receptors (NLRs). NLR activation drives the assembly of an inflammasome consisting of oligomerised NLRs, adaptor apoptosis-associated speck-like protein containing a CARD, and caspase-1. Caspase-1 activation and cleavage enables the cleavage of IL-1β and IL-18 and pore-forming molecule gasdermin D (GSDMD). IL-1 β is released through GSDMD pores, which result in lytic cell death. *M. tuberculosis* infected THP-1 macrophages caused the formation of ASC specks and were only localised in dead cells, and IL-1 β secretion increased [121]. The secretory antigenic target secretion system 1, ESX-1, mediated plasma membrane damage causing K^+^ efflux, NLRP3 activation, and subsequent caspase-1 mediated IL-1β release in THP-1 cells.

A subset of *M. tuberculosis* infection can occur in non-myeloid cells. *M. tuberculosis* with functional type VII secretion systems and Phthiocerol dimycocerosates (PDIM) form intracellular cording phenotypes that have been reported within the human lymphatic endothelial cells (hLEC) [122]. *M. tuberculosis*-infected hLEC gain access to the cytosol, whereby cording occurs. The bacilli prevent the activation of host-immune sensors, thus preventing xenophagy and the persistence of chronic infection [122].

Anti-tuberculosis drugs have been shown to be less effective in vitro against non-replicating mycobacteria generated under hypoxic conditions [123]. An important advance for our appreciation of the complexity within granuloma was achieved by dual staining techniques that confirmed the presence of heterogenous sub-populations of *M. tuberculosis* in hypoxic cultures and lung sections [123,124]. These sub-populations exhibited differential susceptibility to antimicrobial agents and required differential growth requirements in vitro [125]. Culture supernatant from actively growing *M. tuberculosis* contains a family of cell wall cleaving enzymes known as resuscitation promoting factors (Rpfs) involved in the resuscitation of differentially culturable bacteria [126,127]. Together, these cells can persist and cause relapse in patients and thus cause a state of chronic infection. Persister cells are defined in two ways: firstly, those cells that evade the antibiotic but are drug-susceptible upon subculture, and secondly, bacteria that can switch to being metabolically inactive and thus exhibit persistence through slow growth or none at all [128].

## 8. Anti-Tuberculosis Drugs and Immune Modulation

The anti-tuberculosis drugs currently available for the treatment of tuberculosis not only render the bacilli defective through unique mechanisms, they also have some effect on the immune response. This section focuses on the interplay between the anti-tuberculosis drugs and the immune response.

Rifampicin (RIF) is one of the most potent anti-tuberculosis drug and was introduced into the anti-tuberculosis drug regimen in 1968. RIF inhibits *M. tuberculosis* by binding to the β subunit of DNA-dependent RNA polymerase; resistance is associated with the *rpoB* gene. This drug has been shown to increase CD1b expression, which is found on cytokine-activated macrophages, thus boosting the T-cell response to *M. tuberculosis* [129]. Additionally, RIF was shown to augment NO production in human alveolar epithelial cells, which are activated by IL-1β, TNF-α, and IFN-γ [130]. Prostaglandin E_2_ (PGE_2_) is involved in the activation of antigen presenting cells, regulation of T- and B- cell responses, and chronic inflammation. PGE2 production was inhibited by RIF in human alveolar cells [131]. RIF inhibits prostaglandin E2 production and arachidonic acid release in human alveolar epithelial cells. Others have reported rifampicin to exert no effect on the production of NO but prevented phagocytosis in mouse macrophages [132]. RIF and dexamethasone have similar effects on the macrophage phagocytosis of zymosan but differ in their effects on nitrite and TNF-alpha production.

Isoniazid (INH) was approved by the FDA in 1952 for treatment against tuberculosis. INH is oxidised by the *M. tuberculosis* catalase-peroxidase KatG, which produces free radicals and in turn inhibits mycolic acid synthesis [133,134]. Phagocytic cells produce ROS/NOS after infection with *M. tuberculosis* to clear the bacteria; however, *M. tuberculosis* is highly resistant to ROS/NOS, and thus the phagocytic cells undergo necrosis. A study has shown INH can inhibit the oxidative stress-induced necrosis of phagocytic cells; however, this is not fully understood [135]. Additionally, INH induces the apoptosis of CD4^+^ T cells, autophagy, and phagosomal maturation in response to *M. tuberculosis* [136,137].

Pyrazinamide (PZA) has been recognised as an anti-tuberculosis drug since 1952. PZA is converted to pyrazinoic acid by nicotinamidase encoded by *pncA* and is only activate against bacilli in an acidic pH or inside macrophages [138]. This drug reportedly reduces the release of pro-inflammatory cytokines such as IL-1β, IL-6, and TNF-α from *M. tuberculosis* infected monocytes [139].

Bedaquiline (BDQ) is a recent FDA-approved drug used for the treatment of multi-drug resistant tuberculosis (MDR-TB). This drug inactivates bacterial ATP synthase and thus depletes the pathogen of ATP [140]. BDQ reportedly triggers phagosome-lysosome fusion and autophagy in macrophages infected with *M. tuberculosis* [141,142].

Clofazimine (CFZ) demonstrated anti-tuberculosis action in the 1950s; however, extreme side effects were reported—for example, skin discolouration and mental disturbances. Thus, its use was halted. However, in the early 1960s, CFZ was approved again by WHO for the treatment of drug-resistant M. leprosy. Additionally, the emergence of extremely drug-resistant *M. tuberculosis* (XDR-MTB) has shown interest in CFZ once again [141]. CFZ is enzymatically reduced followed by spontaneous oxidation and results in the production of ROS [143]. The potassium channel Kv1.3 has been associated with the activation and function of T lymphocytes. CFZ has been reported to inhibit the T-cell receptor signalling pathway by blocking IL-2 production and thus results in immunosuppressive activity [144].

## 9. Conclusions

The *M. tuberculosis* research field has mainly focussed on the impact of complement activation on the actively growing mycobacterial populations; very little is known about the dormant, non-replicating, and resuscitation of heterogenous sub-populations of mycobacteria that exist within granulomas.

To understand the complement cascade in *M. tuberculosis* infection, we need to continue advancing in our knowledge of the infection and immune response to *M. tuberculosis*. To date, we carry out macrophage infections in vitro using *M. tuberculosis* but do not understand fundamental questions. Why are macrophages not infected equally whilst using a homogenous *M. tuberculosis* culture and immortalised cell lines? Why does multiplicity of infection need to be tightly controlled in vitro, and why do a large number of bacteria overwhelm the macrophage if there are only a limited number of receptors for phagocytosis?

All three complement pathways together play a role in the pathogenesis of *M. tuberculosis* infection; although poorly understand, this review sheds light on the complexity of tuberculosis infection.

## Figures and Tables

**Figure 2 medicina-57-00084-f002:**
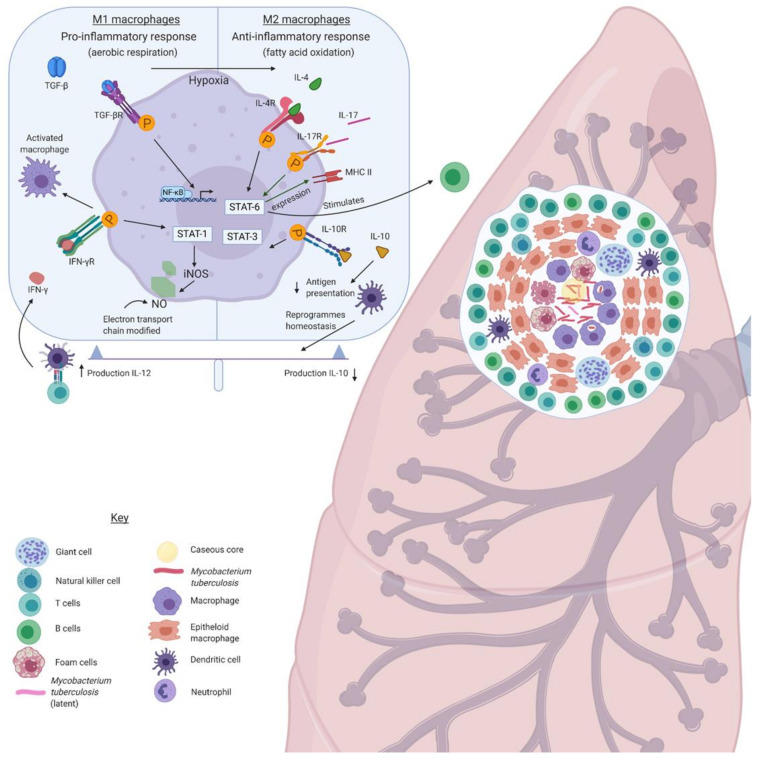
Overview of the *M. tuberculosis* granuloma.

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
