# Peer review of "The Role of Complement System and the Immune Response to Tuberculosis Infection"

_medicina, 2021, doi:10.3390/medicina57020084_

Round 1

Reviewer 1 Report

This is well written and comprehensive review on the role of complement and innate  immunity on Mycobacteria infections. As stated in the introduction, the review focuses on the role of complement, since this topic has not been covered to much extent in previous work and reviews. However, although the title is The role of complement in tuberculosis infection, the review also cover many other aspects of immune regulation during mycobacteria infection, as well as some aspects of antibiotics. Therefore, the review should be edited to focus on the role of complement, since the other aspects have been covered in several previous reviews.

Minor points

line 144-46: The role of TSR4-5 is unclear to me. Is it stimulating or impairing phagocytosis?

line 311: histocytes should read histiocytes

line 371: ref 72 does not discuss the Th1/Th2 balance

line 470:Rifampicin augment NO production is stated twice.

Author Response

Thank you reviewer 1 for all of your comments.

Comments:

We agree that the review focuses on the role of the complement system during tuberculosis infection. We added more details on the overall immune response to tuberculosis, because we felt that the downstream immune responses are an indirect result of the complement system activation. We therefore wanted to show how important the complement system is, and how understanding the complement system could shed light on the early phase of infection. I think that a change in title from 'The role of complement in tuberculosis infection' to 'The role of the complement system and the immune response to tuberculosis infection' maybe better suited.

Minor points:

line 144-46: The role of TSR4-5 is unclear to me. Is it stimulating or impairing phagocytosis?

We have removed these lines to provide clarity on the role of TSR4-5 which is to recruit and stimulate phagocytosis

line 311: histocytes should read histiocytes

change made

line 371: ref 72 does not discuss the Th1/Th2 balance

this should be reference 148

line 470:Rifampicin augment NO production is stated twice.

Removed duplication

Reviewer 2 Report

This is a well written narrative review on the role of the complement System in the immune Response against M. tuberculosis.

The review is excellent, with an in Depth summary of the present state of the Knowledge, Wonderful and clear figures, and an embracing Content.

As a minor issue one or two further figures would be fine as especially the first few pages would Benefit from a figure.

Author Response

Thank you reviewer 2 for all of your comments.

We really appreciate the feedback for the figures. We feel that adding more figures to the first few pages of the review would add no further benefit to the review. The first few pages include an introduction and the accepted paradigm of infection for tuberculosis. By adding a figure for these sections, would only be replicating figures that have already been published. We are happy to generate a specific figure if you feel this is necessary.